# Measurement and Control of Corrugated Board Production Parameters Taking into Account Individual Operator Preferences

**DOI:** 10.3390/s23146478

**Published:** 2023-07-18

**Authors:** Paweł Pełczyński, Krzysztof Kadys, Włodzimierz Szewczyk

**Affiliations:** 1Centre of Papermaking and Printing, Lodz University of Technology, Wólczańska 221, 95-003 Lodz, Poland; wlodzimierz.szewczyk@p.lodz.pl; 2Rawibox S.A., Podmiejska St. 14, 63-900 Rawicz, Poland; krzysztof.kadys@rawibox.com.pl

**Keywords:** corrugated board, process variable measurement, experience of operator, production parameters

## Abstract

The article presents a proposal for optimizing the production process of corrugated cardboard based on measurements of process variables as well as the knowledge and skills of the operator conducting production. This technique involves continuous recording and analysis of process quantities that affect the quality of the produced cardboard. For this purpose, a network of temperature sensors with a system of continuous registration and monitoring of the process variables was designed and installed in the industrial environment of the corrugator. The recorded data is analyzed to estimate the desired values of the measured process variables, giving clues to how to control the production line. Unlike existing systems, the proposed algorithm for controlling production parameters allows each operator to use their preferred values for process variables independently of others. The proposed system allows for improving the quality of the produced cardboard and increasing the efficiency of its production by taking into account the individual experience and habits of the operator conducting production.

## 1. Introduction

Corrugated cardboard is a well-known material used for the production of packaging [1,2]. Their usefulness is determined by the strength properties of cardboard. The high strength of the cardboard allows for minimizing the weight of the packaging while maintaining the required strength [3,4]. Therefore, during the production of corrugated cardboard, great emphasis is placed on its mechanical properties, which determine the properties of packaging produced from it, in particular collective packaging, which is stored in high stacks [5,6,7,8]. Achieving good strength with a low packaging weight is also economically justified and important from the point of view of minimizing the consumption of natural resources [9]. Packaging products in the retail trade in cardboard packaging is also associated with the need to achieve high aesthetic values. They result mainly from the way the packaging surface is finished and printed, but also from the surface quality of the cardboard used [10]. The quality of the surface of cardboard is determined both by the quality of the papers used for its production [11,12,13,14] and the corrugated board manufacturing process. Hence, it is increasingly important to be able to control the quality of manufactured cardboard during production and quickly remove the causes of potential product defects.

Currently, automatic product quality control and management systems are not widely used in the production of corrugated board. One of the most advanced solutions is the WCS (Warp Control System) from BHS [15]. The system consists of several cooperating software components, temperature sensors, and devices controlling the production process installed in the technological line. Due to the need for continuous temperature measurement of papers used in the production of cardboard, directly before the glue units called single facer and double facer or double backer, and the need to measure the temperature of cardboard webs and the final product (dobuleface board), stationary pyrometers have been employed as temperature sensors, which is a common practice in the industry [16,17]. In order to regulate the temperature of the raw materials, the system controls the wrapping angle of the heating cylinders and the pressure of the steam supplying these cylinders. The system operates in a closed-loop temperature regulation mode, allowing for the elimination of regulation error and precise adjustment of the raw material temperature [18]. As a result, the risk of problems with maintaining the flatness of the produced cardboard sheets is reduced [19,20]. The system retrieves information about production parameters from the planning system and takes it into account when setting the parameters of the production process. Knowledge about the most desirable selection of paper temperatures is developed based on the registration of settings applied by the corrugator operators during previous productions of the same cardboard. It is assumed that adjustments made by the operators to the initial production parameters lead to improvements in selected product quality characteristics and should be considered when setting the initial parameters for future productions.

Another quality control system is the QCS system currently being developed at Val-Met in cooperation with Saica [21]. Its task is to automatically control the temperature and moisture of raw materials based on the measurement of geometrical parameters of the product. For this purpose, the shape of the cross-sectional profile of cardboard sheets is evaluated, and the transverse temperatures and humidity profiles of the board-produced cardboard are also measured. Based on the real-time recorded data, it is possible to evaluate the transverse moisture profile of the cardboard webs and the paper for the bottom flat layer before the double facer using electronically controlled humidification beams.

Current quality management systems for manufactured cardboard [22,23] are characterized by the ability to learn from historical data, taking into account settings or their corrections made by a human, but none of the existing systems uses the individual experience of a given operator to optimize process quantities according to their preferences. As a result, attempts by different operators to apply the same initial settings to a production range usually end up with new settings that are best for them individually. Such behavior causes transitional states, in which the quality of the product may differ from the desired one. In the case of short production cycles, this can cause an overall reduction in the quality of the produced board [20]. The need to avoid these problems and to use the potential of the operator’s individual habits became the motivation to conduct the presented research. The aim of the research was to develop a proposal for an algorithm optimizing the corrugated board production process based on the knowledge and skills of operators. For this purpose, a system for continuous recording of process variables was designed and installed in the corrugator, and the recorded data were analyzed in terms of the differentiation of their values depending on the operator managing the production line. Based on the analysis, an algorithm for optimizing the production parameters of corrugated board was developed.

## 2. Materials and Methods

### 2.1. Design of a System for Continuous Measurement and Recording Process Variables

One of the very important quality indicators of the corrugated board produced is the flatness of the cardboard sheets. It directly depends on the distribution of moisture content in the raw materials and webs of single-face (double-layer) cardboard produced in single facers. Measuring the moisture content of corrugated board is difficult, and usually the measurement results are subject to large errors [24]. It should be noted that the moisture content is influenced by the temperature distribution in the raw materials and in the single-face board webs. Both the measurement and control of temperature can be accomplished using well-known techniques [25].

The implementation of the control of corrugated board production parameters [26], taking into account the individual preferences of the operator, required installing and configuring a network of process variable sensors with a recording system and software for their analysis. The development of the discussed system was part of a larger project aimed at introducing process innovation in adhesive preparation and dosing, where a local circulation of adhesive with control of its concentration directly at the glue pans was implemented. The work began with a series of measurements of process variables using portable thermometers and moisture meters [27,28,29]. Based on the analysis of temperature and moisture measurements of the raw materials and the product, measurement points were selected for monitoring using a continuous registration system for process variables [30]. It was determined that the temperature and moisture content of the raw materials before heating had no significant effect on the quality of the product. Moisture measurement was abandoned, and the number of temperature measurement points was limited to the points that were most important for the quality of the product. Next, the system was installed in the technological line of the corrugator and configured. Continuous measurement of the surface temperature of paper or cardboard requires the use of non-contact measurement techniques [31,32,33]. The system consists of pyrometers and immersion thermometers, a corrugator speed transducer, and recorders of measured values connected to a local computer network. It was decided to use stationary pyrometers manufactured by Optris GmbH, Germany [34].

The following temperature measurements were made:Temperature of flutes (papers for corrugated layer) and liners (papers for flat layer) at the facer’s inputs and cardboard webs at the facer’s outputs,Cardboard temperature in the drying section, after the drying table, and after the pulling section,Adhesive temperature at the inlet and outlet of glue pans.

After installing the continuous temperature recording system at selected points in the technological line, several series of measurements with portable instruments were made in order to verify the correct operation of the continuous recording system of process quantities [35].

### 2.2. Selecting Measurement Points in the Cardboard Production Line

In single facers, it is important to control the temperature of the papers after heating and conditioning, the temperature of the web of glued single-face cardboard, and the temperature of the adhesive entering the glue pan and at the overflow from the pan [36]. The temperature of the papers at the entrance to the facer determines the gluing process of the cardboard layers and, as a result, the gluing strength. The temperature of the resulting double-layer board determines the rate of moisture removal, which affects the flatness of the produced cardboard sheets. The temperature of the glue in the glue pans must not be too high to prevent gelation, which would adversely affect the properties of the glue. The measuring points in both single facers of the technological line are shown in Figure 1. In contrast to the standard temperature control systems for the webs (points P3P2 and P5P4), it was also decided to measure the temperature of the papers (points P2b, P3c, P4b, and P5c), which allowed assessing the correctness of their heating before entering the single facers. In addition, measurement of the temperature of the glue at the inlet and outlet of the glue tanks was introduced. Immersion thermometers were used for this purpose.

Table 1 describes the meaning of each symbol and shows the range of measured temperatures.

In a double facer, it is desirable to monitor both the temperature of the adhesive entering the glue pans and at the outflows from these pans, as well as the temperature of the webs of the single-face cardboard and the paper of the covering layer at the exit of the doble facer. Similarly, it is very important for monitoring the entire corrugated board production process to measure the temperature of the board coming out of the drying table and at the exit of the drying section. The location of the measuring points in the double facer and dryer assembly is shown in Figure 2 and described in Table 2.

### 2.3. Architecture of the Measurement System

The system of continuous temperature measurement at selected points of the technological line has been built and configured in accordance with the plan of location of measurement points shown in Figure 1 and Figure 2. In addition, the instantaneous speed of the board of the produced cardboard was recorded. The block diagram of the system is shown in Figure 3. The measurement system consists of an air blowing installation protecting the optics of pyrometers, a network of temperature sensors located at the points of the production line, and recorders of measured values. The recorders were connected to a local computer network, which allowed reading the recorded signals using dedicated software installed on a computer connected to the same local network. The measuring system was built using sixteen channel recorders from APAR (Raszyn, Poland) of standard 0–10 V voltage signals of type AR207/16U/S1/P/P/P/P/IP30.

A set of meters has been connected to the recorders according to the description of the measuring points in Figure 1 and Figure 2:W1a, W1b, W2a, W2b, W3a, W3b, W4a, W4b—immersion thermometer AP-TOPGN1-100-Pt100-M20x1.5-0-70C from Termoprodukt (Bielawa, Poland);P5c, P4b, P3c, P2b, P1b, P3P2c, P5P4c, TdG, TdD— pyrometer OPT CS LT15 SF CB3,P5P4, P5P4—pyrometer OPT CS LT15 SF CB8;TcSO—OP CT LT15 pyrometer installed on the operating side;TcSr—OP CT LT15 CB3 pyrometer installed in the middle;TcSN—OP CT LT15 CB3 pyrometer installed on the drive side;Speed—galvanic separator of analog voltage signal 0/1–10 V to 0/1–10 V SEPGAL U/U for recording the signal of the speed of travel of the board produced in cardboard.

Figure 4 shows examples of non-contact temperature measurement points using pyrometers. The sensors were installed on metal racks connected to the supporting structure of the corrugator in such a way that it was possible to adjust their distance from paper and cardboard webs.

Figure 5 shows the glue temperature measurement points in the glue pan of the first single facer. Immersion thermometers with Pt100 sensors were installed in them. The method of their installation allowed for precise determination of the temperature of the glue flowing into and out of the glue pan.

In order to protect the optics of pyrometers from dust during operation, they were placed in covers to which compressed air was supplied [37]. As a result, the air stream in the vicinity of the pyrometer lens prevents dust from settling on its surface. The intensity of blowing has been individually selected for a given measuring point using local valves on compressed air supply hoses.

Registration of the moment when a good quality product is achieved is done by pressing a button installed at the operator’s station. The following factors are taken into account to assess the quality of cardboard during production: assessment of the flatness of the lying of cardboard sheets; assessment of the tear resistance of cardboard layers; measurement of the dimensions of the sheets produced; measurement of cardboard thickness; and assessment of the adhesive joint.

## 3. Results

### 3.1. Presentation and Discussion of the Results of Continuous Measurements of Process Data 

The following software has been developed by the authors to analyze the results of continuous measurements:Python program for selecting and combining measurement data from two recorders into one array of time-synchronized values from a selected day (developed in the official Python 3.7 Idle environment);Software for averaging and merging daily measurement data into one set enriched with data from the production history file (also the Python program);VBA macros to visualize the data prepared in this way (developed in the MS Office 365 environment).

The analysis was performed on a computer running the Windows 10 operating system, equipped with an i7 processor and 8GB of RAM. The analysis did not require much computing power.

Examples of diagrams of process quantities as a function of time are shown in Figure 6, Figure 7 and Figure 8. The diagrams cover the period of continuous production, during which, in order to verify the values of the measured quantities, measurements of process quantities were also made with portable instruments. In all charts, changes in machine speed are plotted in black. This gives the opportunity to find the moments of change in the manufactured assortment and link the values of the measured temperatures with the production of a given type of cardboard. 

Most graphs exhibit significant speed fluctuations during the initial phase of a production cycle when changing the produced assortment. This is also followed by fluctuations in the surface temperature of papers and cardboard in some cases. After initial fluctuations, the temperature values stabilize at a level that is maintained until the end of the given production. Additionally, temporary speed drops occur during the production of a given assortment. They are associated with the exchange of paper rolls.

The temperature waveforms of papers and cardboard webs in the second single facer are shown in Figure 6. They are not subject to strong fluctuations despite multiple changes in the cardboard assortment produced. The highest temperature is reached by paper for a flat layer. The web of produced cardboard has the lowest temperature. It is caused by the drop in temperature as a result of contact with the adhesive at a lower temperature and the evaporation of water contained in the adhesive. The temperature waveforms of the lower layer of cardboard after the drying table shown in Figure 7 reveal the unevenness of the transverse temperature profile, which may affect the flatness of the produced cardboard. The temperature waveforms of the upper and lower layers of the cardboard after the dryer shown in Figure 8 show the difference due to the heating of the lower layer of the board on the drying table. Figure 9 shows the temperature waveforms of the adhesive at the inflow and outflow from the glue pan of single facer 1.

Based on the analysis of the recorded temperature waveforms at each measuring point, several conclusions were drawn regarding the production process. The temperature of the upper surface of the cardboard plate downstream of the dryer (TdG point) is lower than the temperature of the lower surface by a constant value, depending on the type of cardboard produced. In the case of five-layer cardboard, this difference is the largest—at the level of 12 °C to 20 °C. In the case of a three-layer B-wave board, it is also usually maintained at a high level of 10 °C to 15 °C. Slightly smaller differences occur in the case of C-wave cardboard. The temperatures recorded at TcSO, TcSr, and TcSN should be the same, as they represent the transverse temperature profile of the lower surface of the cardboard. The observed differences are a result of uneven heat flux transferred over the width of the dryer, caused, for example, by the uneven pressure of the cardboard on the boards of this table. These differences may cause later problems with the flatness of the sheets of the produced cardboard, which is an important factor determining their quality. Therefore, the measurement and recording of the transverse temperature distribution of cardboard is very desirable from the point of view of maintaining good product quality.

The adhesive temperature in the single facer 1 bath in the analyzed time period remained at a level not exceeding 30 °C. This value is definitely lower than the gelatinization temperature, which means that there is no danger of losing the required adhesive properties before joining the layers in the cardboard web. In addition (Figure 9), a slight heating of the adhesive in the bath can be observed as a result of contact with the cylinder of the adhesive unit. The temperature difference between the inflow and outflow most of the time does not exceed 1 °C.

### 3.2. Results of Measurements of Process Variables during the Manufacturing of Cardboard Produced in the Analyzed Period 

Based on the analysis of the production history report in November and December 2021, the most frequently produced cardboard with the designation CB67-079N was selected, and an analysis of the temperature distribution at continuous recording points during subsequent productions was performed on it, depending on the individual preferences of the settings by the operator conducting production. This type of cardboard was produced 90 times over a period of 2 months, which allowed us to obtain statistically significant comparison results. The average temperature values in the analyzed period at the individual measuring points and the average speeds used by each operator are shown in Table 3.

Progressive averaging was applied to temperature and velocity values, which involves taking the average of all values achieved in productions from the first one in the analyzed period up to the current one. Figure 10, Figure 11, Figure 12 and Figure 13 show the results of averaging temperatures from the last 50 productions at selected measuring points. These include both values for production by each of the three operators separately and values for all operators. Measurements of individual process quantities were carried out in the absence of knowledge of operators about the history of changes in these quantities during production carried out by both a given operator and other operators. This made it possible to assess the difference in the selection of process parameters by individual operators in subsequent productions of the same assortment.

## 4. Discussion—Design of the Algorithm for the Selection of the Desired Values of Process Quantities Considering the Individual Preferences of the Operator

### 4.1. Justification of the Need for Individualized Optimization of Process Parameters

The quality of corrugated cardboard is influenced by both the quality of the raw materials and the technological process of its production. In the technological line of the corrugator, it is possible to control this process by setting its parameters from the control panel. In the control systems of modern corrugators, the desired temperature values of raw materials are set, and the automation system controls the adjustable quantities so as to achieve the set parameter values. Regardless of the automatic setting of selected process parameters, the operator has the ability to change them.

In the examined technological line, dependencies of recorded process quantities on the individual habits of the operator conducting production were observed. Temperature distributions at individual measuring points averaged progressively over a two-month period tended to reach fixed values. At points P3P2, P2b, P5P4c, and P3P2c, temperature distributions tended to a single value, independent of the operator. At some measuring points, however, a different behavior of the recorded temperature waveforms was observed. For example, the temperature waveforms at points P5P4 and P4b shown in Figure 10 and Figure 11 tend to have a different value in each operator. Since the recorded data relate to the same production conditions, the observed differences suggest the influence of the individual preferences of the operator conducting production on the setting of its parameters. The differences may result from the fact that each operator adopted a different goal and had in mind different product parameters that determined its quality. For one operator, the most important determinant may be efficiency measured by machine speed, while another may pay more attention to maintaining the best possible flatness of the sheets of produced cardboard or the greatest possible energy savings. It is difficult to clearly assess what goal should be achieved in the first place. However, it can be seen that the sets of process parameters given by each operator tend to a constant value as they are averaged, which means achieving the optimum in the subjective understanding of the goal function. It is possible to automate the setting of initial production parameters for a given type of cardboard, either individually for a given operator according to their previous experience, or averaged over all operators, which is a compromise between individual preferences for settings. However, this requires knowledge of the values of individual process variables and, consequently, their continuous measurement. It is desirable that the set parameters be achieved by automatic adjustment of machine settings, e.g., angles of wrapping of heating cylinders by paper or feed steam stream. In the studied technological line, there is no possibility of automatic adjustment of process variables. On the other hand, a visual presentation of desired and currently achieved values can be a valuable indication for the operator.

An additional argument justifying the need for continuous monitoring of process quantities is the possibility of quick detection of emergency situations or the occurrence of problems in the technological line that deteriorate production conditions, e.g., watering of cylinders or heating plates, or uneven web tension. Detection of deviations of process variables from typical for a given production is a good indicator of these situations and allows for quick reaction, often protecting against the production of batches of cardboard with worse properties.

### 4.2. Algorithm for Optimizing Production Parameters

The essence of the algorithm is to provide the operator with a tool that allows them to optimize the parameters of the production process in accordance with their own experience, enabling the use of data from previously used settings. For this purpose, continuous measurement of the surface temperature of the papers used for production, the webs of the produced single-face board, the board produced, the temperature of the adhesive in the glue pans, and the speed of the machine should be continuously measured in the process line of the corrugator. The measured values are recorded by the measurement data acquisition system, then they are processed in accordance with the developed data analysis algorithm and presented to the operator on an ongoing basis as the desired values during the production of the selected cardboard assortment.

The data analysis algorithm is based on averaging the values of measured quantities from the last several dozen productions carried out by a given operator (the number of productions is a parameter of the algorithm) and for a given type of cardboard. The implementation of the algorithm consists of the following actions:Entering an individual database of paperboard whose production starts if it has not been previously produced by the operator concerned, establishing and recording in the database the initial production parameters in accordance with the operator’s experience in the production of similar paperboard or the experience of another operator in the production of the board in question, if it has already been produced;Taking from the database the average value of previously recorded parameters of a certain number of cardboard productions and taking them as the initial parameters of a given production;Modification of set parameters to improve board quality or process efficiency under given conditions;Adding modified parameters to the database with automatic removal of the oldest production parameters if the number of production data exceeds the assumed history size.

The operation of the main data processing loop within the proposed algorithm is shown in Figure 14.

A comparison of the averaging results of the flute temperature distribution in a B-wave single facer during CB67-079N board production by three different operators shown in Figure 11 reveals a discrepancy in the settings used by individual operators. A similar effect can be seen in Figure 12, which presents a comparison of the results of averaging the temperature distribution of the liner in a single facer producing wave C. Small discrepancies in the recorded temperatures are also visible at the P3P2c point, i.e., on the surface of the flat layer of one of the cardboard webs after the double facer (Figure 13). In the last two cases, the observed differences are at the level of one degree Celsius, which in the case of individual measurements could be treated as an accidental error; however, the fact of averaging a large number of measurement data improves the accuracy of the measurement and allows us to assess the actual differences in temperature values during production carried out by different operators. In other places, there are also discrepancies between the average values of temperatures achieved by all operators and the values obtained by each operator, although these differences are smaller.

Different settings are used by each operator in an attempt to achieve different goals in the production process. For example, the combination of machine speed (the transport speed of the cardboard produced) shown in Figure 15 reveals the desire of operator No. 1 to achieve the highest possible production efficiency. The average machine speed in the analyzed period was 140.5 m/min for the first operator, while the average for other operators was 128.7 m/min. The higher speed used by operator 1 did not cause any deterioration in the mechanical properties of the produced cardboard, as evidenced by the results of the bursting strength and Flat Crush Test (FCT) measurements summarized in Table 4 and Table 5. However, differences in the thickness of the cardboard produced by different operators can be seen (Table 6). The average thickness of the board for operator 1 is 5.82 mm, while the thickness of the board produced by other operators is 6.42 mm. One of the possible reasons for these differences is the viscoelasticity of the paper, whose permanent deformations, and thus the geometrical parameters of the wave, depend on the humidity, temperature, pressure value between corrugating rollers, and duration of this pressure. The behavior of paper exposed to these physical quantities can be described by a four-parameter model, e.g., Burgers [38]. Models with fewer than four parameters are not able to describe the behavior of paper, and, in practice, rheological models are very rarely used to describe its behavior. The gradual increase in speed achieved by all operators in the final production of the analyzed series was the result of the introduction of a process innovation allowing for increased production efficiency.

The results of laboratory measurements of the properties of CB67-079N are presented in Table 4, Table 5 and Table 6.

The bursting strength values of the cardboard produced by individual operators do not show significant differences. In general, the bursting strength of the cardboard is similar to the sum of the bursting strengths of its individual layers, which in the present case did not change. Similarly, Edge Crush Test (ECT) values can be explained, the differences in which in the examined cases are within the limits of measurement errors.

## 5. Conclusions

The main benefit of using the developed system is that it frees the operator from the need to remember the best settings for a given production. Once production starts, the operator has the ability to change process parameters to search for values that provide even better quality or efficiency. Since subsequent achieved process parameters are stored in the system and added to the averaged set, this set of parameters is optimized based on the operator’s assessment. The average history is limited to 50 productions. Older data is not averaged to allow the system to adapt to slow changes in production conditions, e.g., due to changes in ambient humidity at different times of the year.

One of the benefits of using the developed system is that it reduces the risk of errors in controlling the production process. This is achieved by automatically remembering the best settings for a given production. 

Another very important benefit is the successive pursuit of high product quality and efficiency in the production process under the conditions of production carried out by a given operator. The optimization process is then not disturbed by the different habits of different operators in terms of setting the parameters of the production process. This ensures that optimization is not disturbed by the modification of the target function, as is the case when taking into account the production history of many operators. As a result, it is possible to quickly set the operating parameters of the production line for the production of a given assortment of cardboard and, with repetition of the same assortment, find the optimal parameters of its production.

A certain limitation of the proposed technique of optimization of cardboard production parameters is the inability to search for an objectively justified, optimal set of machine settings. However, it would be very difficult due to the multidimensionality of the problem and the large impact of factors independent of the construction of the production line and technological parameters of production.

## 6. Patents

The algorithm for selecting the desired values of process quantities presented in the article, considering the individual preferences of the operator, has been submitted for patent protection—application no. P.442773 “Method of optimizing corrugated board production parameters”.

## Figures and Tables

**Figure 1 sensors-23-06478-f001:**
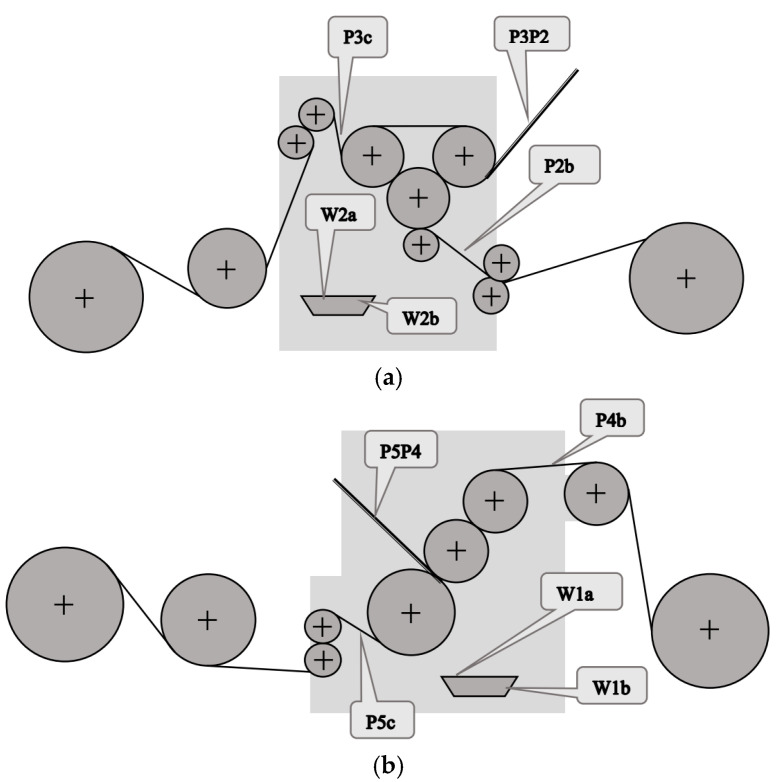
Continuous measurement points (**a**) in the first single facer and (**b**) in the second single facer.

**Figure 2 sensors-23-06478-f002:**
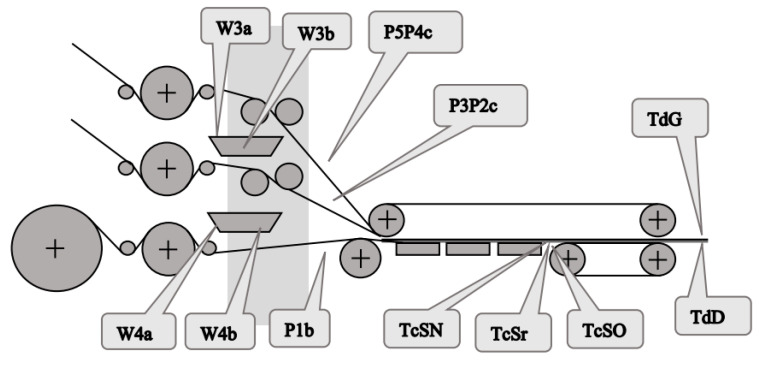
Continuous measurement points in the double facer and dryer assembly.

**Figure 3 sensors-23-06478-f003:**
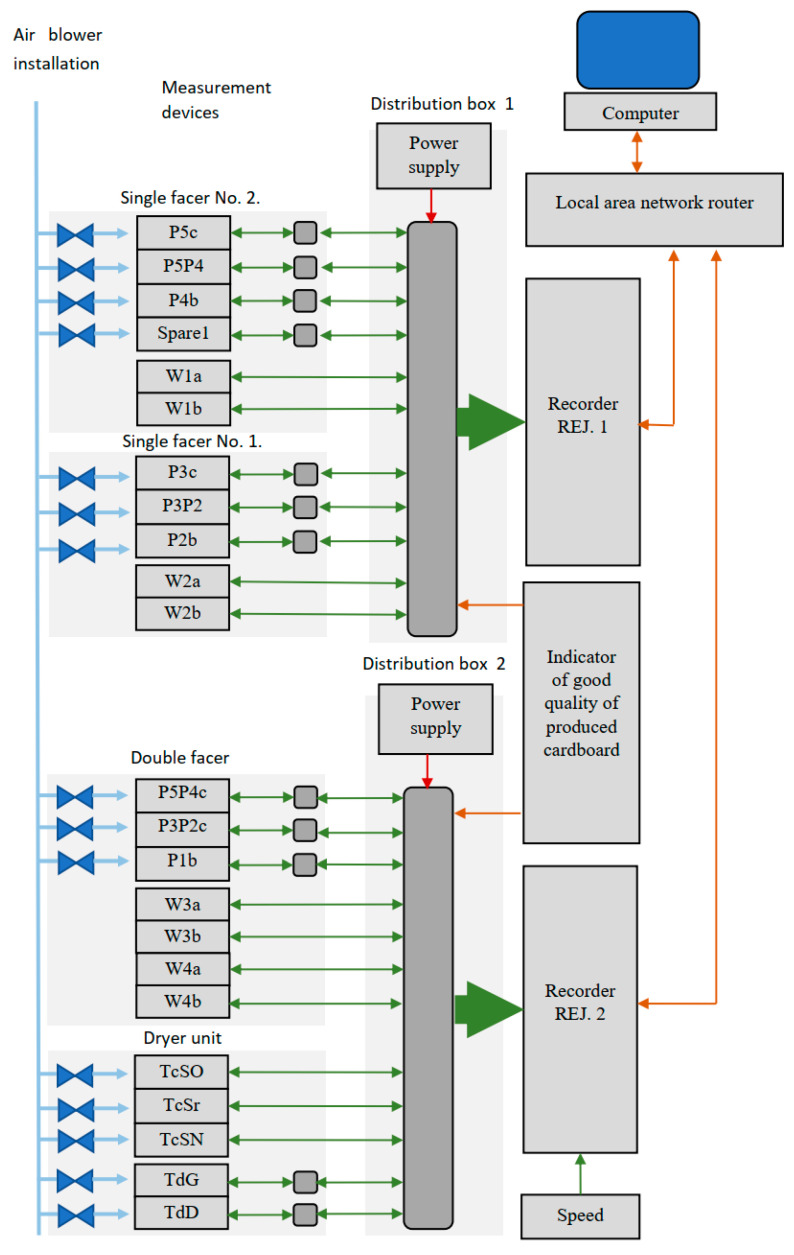
Block diagram of the system of continuous recording of temperatures, speed and determination of good production.

**Figure 4 sensors-23-06478-f004:**
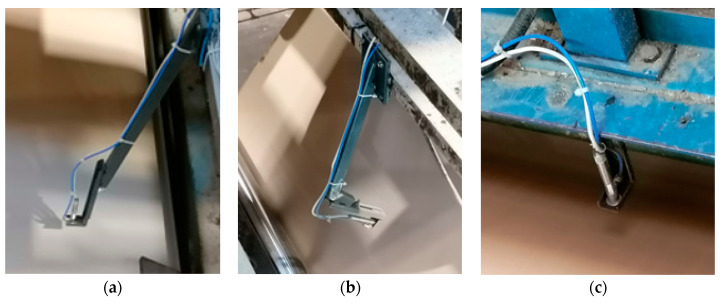
Selected measuring points in the single facer: (**a**) temperature of the paper used for the corrugated layer (flute); (**b**) temperature of the flat surface of the web; (**c**) temperature of the paper used for the flat layers (liner) of the double-layer board.

**Figure 5 sensors-23-06478-f005:**
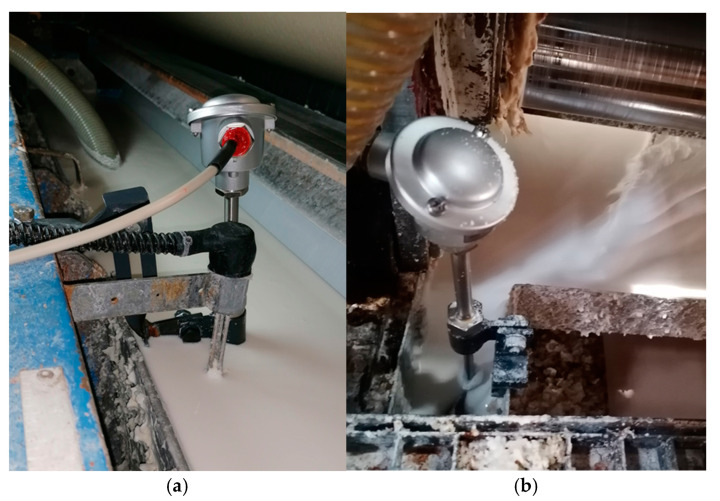
Temperature measurement points in the glue pan of the first single facer: (**a**) on the inlet to the pan; (**b**) on the outlet from the pan.

**Figure 6 sensors-23-06478-f006:**
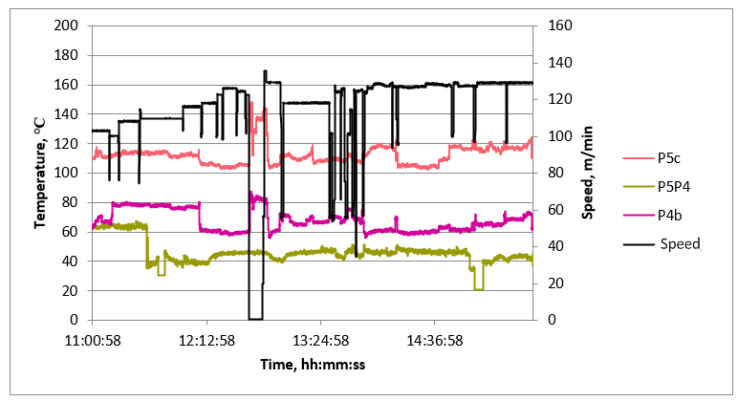
Diagram of machine speed and temperatures at measuring points in single facer 2 as a function of time.

**Figure 7 sensors-23-06478-f007:**
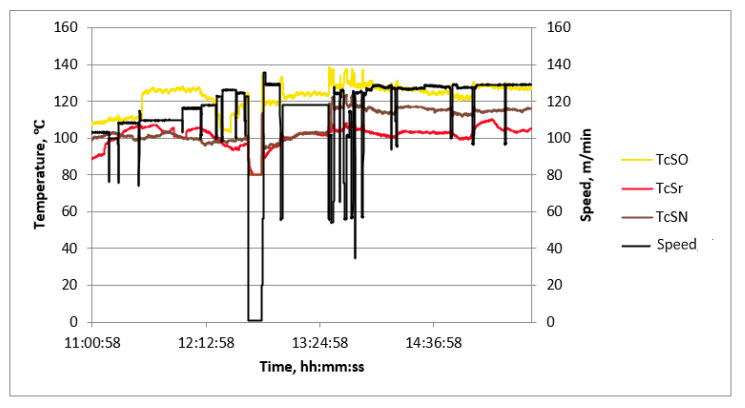
Diagram of machine speed and temperatures at measuring points after the last section of the drying table as a function of time.

**Figure 8 sensors-23-06478-f008:**
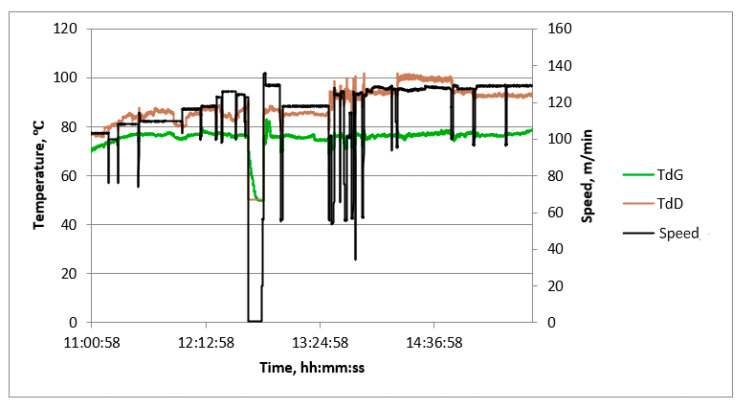
Graph of machine speed and temperatures at measuring points after the dryer as a function of time.

**Figure 9 sensors-23-06478-f009:**
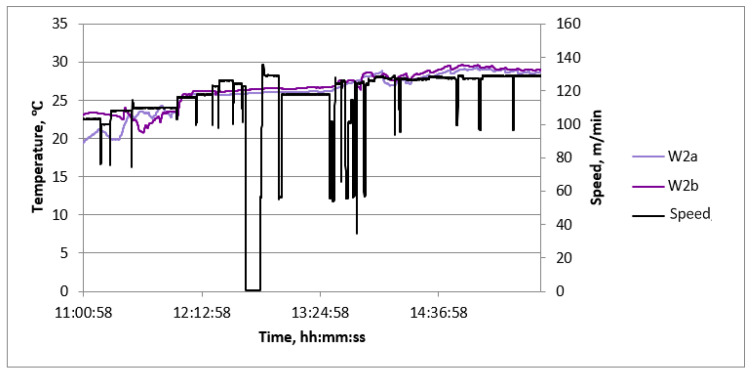
Diagram of machine speed and adhesive temperatures at the inflow and outflow of the adhesive bath in a single facer 1 as a function of time.

**Figure 10 sensors-23-06478-f010:**
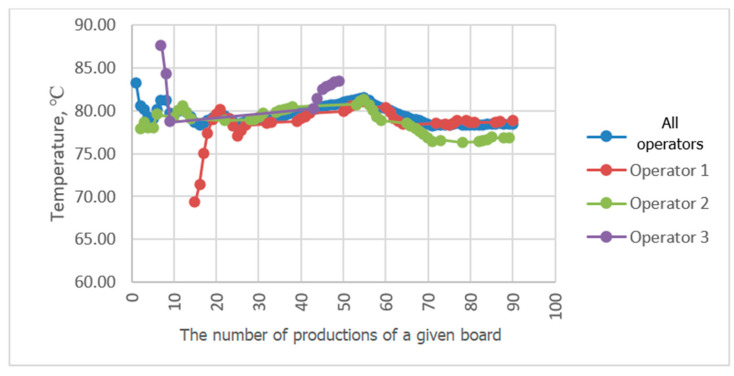
Comparison of P5P4 temperature distribution averaging results during CB67-079N board production by three different operators.

**Figure 11 sensors-23-06478-f011:**
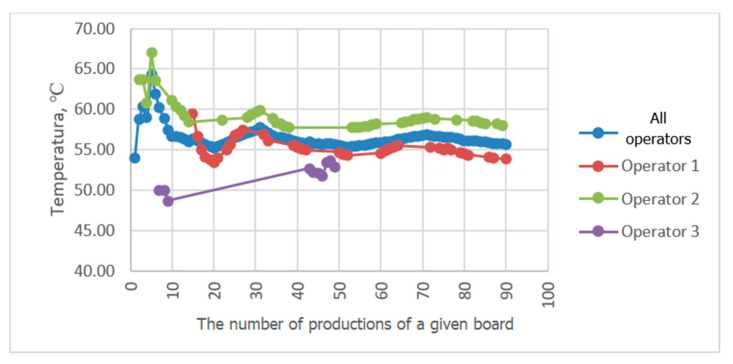
Comparison of the results of averaging the temperature distribution at P4b during the production of CB67-079N cardboard carried out by three different operators.

**Figure 12 sensors-23-06478-f012:**
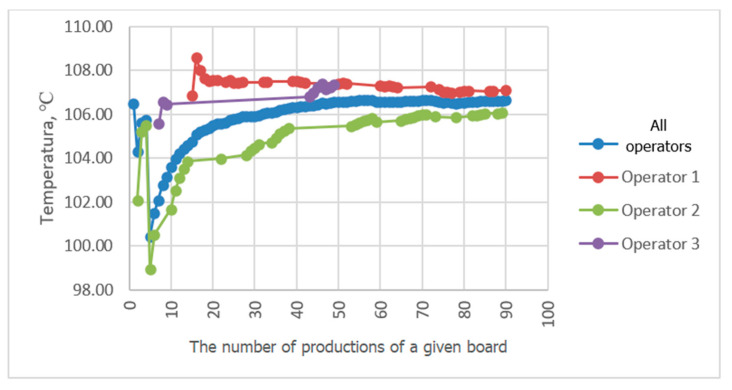
Comparison of P3c temperature distribution averaging results during CB67-079N board production by three different operators.

**Figure 13 sensors-23-06478-f013:**
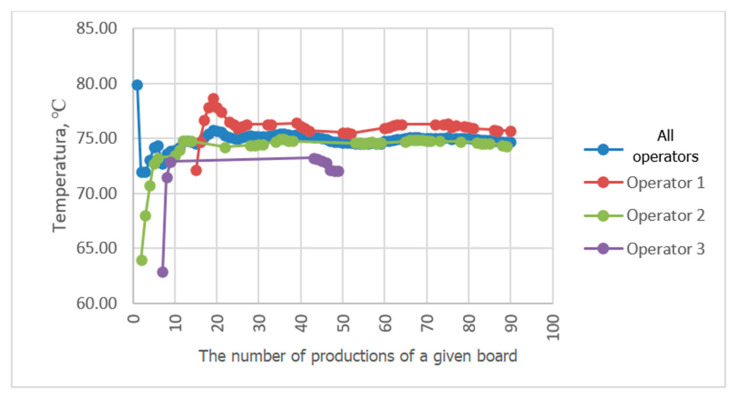
Comparison of P3P2c temperature distribution averaging results during CB67-079N board production by three different operators.

**Figure 14 sensors-23-06478-f014:**
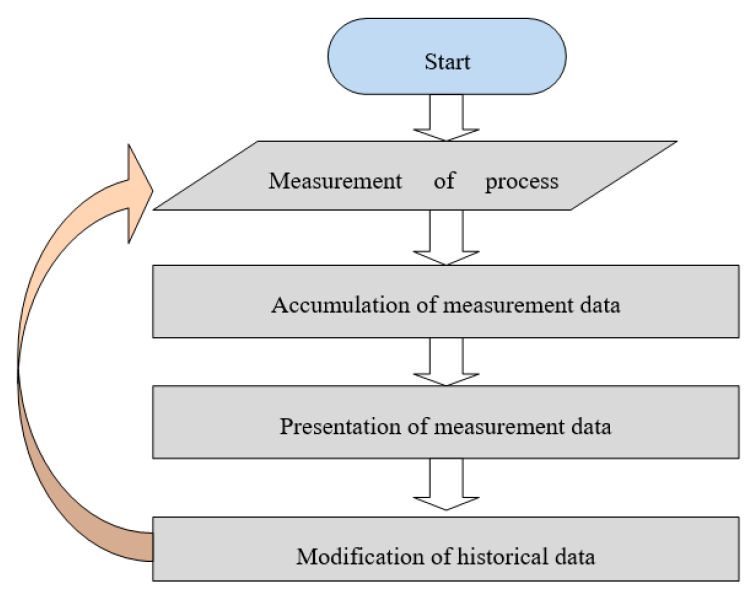
Graphical presentation of the production parameter suggestion algorithm.

**Figure 15 sensors-23-06478-f015:**
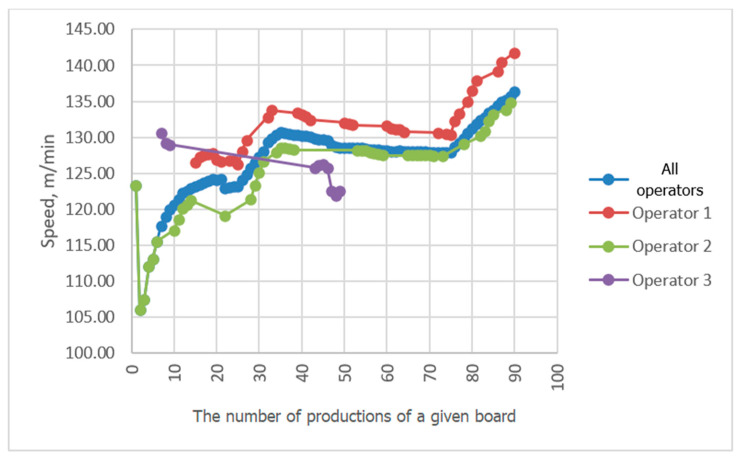
Comparison of cardboard speed averaging results during CB67-079N board production by three different operators.

**Table 1 sensors-23-06478-t001:** Description of continuous measurement points in single facers.

Symbol	Temperature Measuring Object	Temperature Measuring Range
P3c	Paper 3 before entering the pressure belt	50–140 °C
P3P2	Two-layer cardboard at the exit of single-facer No. 2	50–120 °C
P2b	Paper 2 at the output from the preconditioner	50–140 °C
W2a	Glue on the pan inlet	0–70 °C
W2b	Glue on the pan outlet	0–70 °C
P5c	Paper 5 (top layer of 5-ply board) at the input of the corrugating roller	40–150 °C
P5P4	Two-layer cardboard at the exit of single facer No. 1.	20–120 °C
P4b	Paper 4 at the output of the preheater	30–100 °C
W1a	Glue on the pan inlet	0–70 °C
W1b	Glue on the pan outlet	0–70 °C

**Table 2 sensors-23-06478-t002:** Description of continuous measurement points in the double facer and dryer assembly.

Symbol	Temperature Measuring Object	Temperature Measuring Range
W3a	Glue on the inlet to the upper pan	0–70 °C
W3b	Glue on the outlet from the upper pan	0–70 °C
W4a	Glue on the inlet to the lower pan	0–70 °C
W4b	Glue on the outlet from the lower pan	0–70 °C
P1b	Paper 1 before entering the cylinder preceding the heating table	20–100 °C
P3P2c	Bottom web of double-layer cardboard before entering the dryer	50–100 °C
P5P4c	Top web of double-layer cardboard before entering the dryer	50–100 °C
TcSO	Bottom flat layer of cardboard after heating the table on the service side	80–200 °C
TcSr	Bottom flat layer of cardboard after heating the table in the middle of the plate	80–200 °C
TcSN	Bottom flat layer of cardboard after heating the table on the drive side	80–200 °C
TdG	Top flat layer of cardboard after the drying section	50–120 °C
TdD	Bottom flat layer of cardboard after the drying section	50–120 °C

**Table 3 sensors-23-06478-t003:** Average process quantity values for the production of CB67-079N cardboard.

Measurement Point	All Operators	Operator 1	Operator 2	Operator 3
P5c, °C	112.35	110.46	115.17	113.56
P5P4, °C	79.62	79.49	78.54	85.21
P4b, °C	54.76	53.59	57.24	50.91
P3c, °C	106.75	107.22	106.02	107.13
P3P2, °C	93.95	95.31	93.34	87.72
P2b, °C	94.79	95.62	94.82	88.54
P5P4c, °C	64.53	64.36	64.33	66.58
P3P2c, °C	74.58	75.46	74.03	70.34
P1b, °C	50.04	50.10	49.64	50.70
TcSO, °C	114.20	112.50	113.76	124.21
TcSr, °C	101.30	98.96	102.37	107.92
TcSN, °C	105.29	102.51	105.45	118.81
TdG, °C	75.81	75.81	75.65	76.46
TdD, °C	93.31	93.81	92.32	93.94
Speed, m/min	140.49	145.31	137.09	126.69

**Table 4 sensors-23-06478-t004:** Cardboard bursting strength.

Operator	1	2	3	All Operators
Average value, Pa	1475	1483	1489	1479
Standard deviation, Pa	80.5	92.2	143.5	87.7

**Table 5 sensors-23-06478-t005:** Flat Crush Test (FCT).

Operator	1	2	3	All Operators
Average value, N	9569	9600	9675	9591
Standard deviation, N	577	433	247	475

**Table 6 sensors-23-06478-t006:** Cardboard thickness.

Operator	1	2	3	All Operators
Average value, mm	5.82	6.43	6.41	6.11
Standard deviation, mm	0.038	0.052	0.021	0.050

## Data Availability

Data are unavailable due to privacy restrictions.

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
