# Peer review of "Measurement and Control of Corrugated Board Production Parameters Taking into Account Individual Operator Preferences"

_sensors, 2023, doi:10.3390/s23146478_

Round 1
Reviewer 1 Report
Review of sensors-2468723-peer-review-v1: “Measurement and control of corrugated board production parameters taking into account individual operator preferences”
The subject of the paper is relevant with the topics of the journal.
The references are well selected and they are used effectively within the text.
The paper has a good structure and the aims are well described.
It would increase the quality of the paper if the authors were willing to incorporate the following in a revised version of their paper:
· Line 74, the aims and contributions of the paper should be stated clearly
· Line 144, elaborate on figure 3 content
· Line 164, elaborate on figure 4 content
· Line 170, elaborate on figure 5 content
· Figure 10, 11, 12, 13 and 15 correct the X axis legend.
My proposal to the editors is to accept it after minor corrections.
Author Response
The authors would like to thank the Reviewer for valuable comments. Corrections have been made to the text according to the Reviewer's recommendations.
Kind Regards
Pawel Pelczynski
Reviewer 2 Report
Measurement and control of corrugated board production parameters taking into account individual operator preferences
In this study, the authors proposed a method to optimize the corrugated cardboard production process based on the measured process variables as well as the knowledge, and skills of the operator conducting the production. Although the obtained results could potentially be of interest to researchers working on the thematic topic, the paper is not properly organized and would be difficult for researchers to easily follow the paper if it’s published in its present form. To this end, my recommendation is “major revision” because of the following reasons:
Quick comments
1. Are the English language and the style good and easy to understand? No, it should be improved. There is a lot of twisting in the sentences that can and should be corrected.
2. Is the introduction well-written, and concise? No, please check my general comments on the introduction part.
3. Is the paper well-organized and easy to follow? Potentially yes, but an independent section on “Discussion should be provided. Please see my general comments for the discussion section.
4. Are the cited references enough and relevant to the content of the paper? No. The parameters measured should be justified in the introduction and references should be provided to back that up. Also, the shortcomings of what has been done so far are clearly indicated.
General comments
Abstract
(1) The word results in the first sentence of the abstract is not necessary. This sentence can be changed to “This article presents a method to optimize the corrugated cardboard production process based on the process variable measurements as well as the knowledge, and skills of the operator conducting production.”
(2) In the sentence: “For this purpose, a network of temperature sensors with a system of continuous registration and monitoring was designed and installed in the industrial environment of the corrugator.” – What are these sensors registering and monitoring? I believe you wanted to say “…register and monitor the process variables…” If yes, this should be clearly stated and examples of those process variables should be added.
(3) Please write the abstract and correct the English by avoiding unnecessary twisting of the sentences. Also, the measured process variables (i.e., the process variable of interest) should be listed in the abstract.
1. Introduction
(4) In the introduction, please indicate what the study does. You should add another paragraph that clearly explains what this study does and its main contributions in the form of objectives. That is, the aim and objectives of the study should be clearly presented in the introduction.
(5) This introduction is poorly written. The introduction should clearly indicate the importance of this type of monitoring, what has been done and what are the shortcomings of the existing methods, and what you are proposing to address some of the existing problems (the aim of the study), and how you doing it (objectives of the study). I don’t see the aim and objectives of your study in this introduction.
2. Materials and methods
2.1. Design of a system for continuous measurement and recording process variable
(6) Page 2: Line 79: “…a network of process variable sensors…” What are these sensors? – Please indicate what these sensors are and what connection/relationship exists between the process parameters/variables being monitored and the cardboard production process.
(7) Page 2, Line 83-84: In the sentence “The work began with a series of measurements of process variables using portable measuring instruments [25-27].” – What are these instruments? Please indicate. Remember readers should be able to read your paper and understand exactly what you are talking about without going tother references.
(8) Page 2, Line 84-85: In the sentence “Based on the analysis of temperature and moisture measurements of the raw materials and the product, measurement points were selected for monitoring using a continuous registration system for process variables [28].” – What parameter did you consider to determine the accurate position/location of these sensors and why?
(9) In the first paragraph of the subsection, you indicated that you measured the temperature and moisture, but in the list of the measured parameter, you only indicate that you measure temperature at different levels of the process – Please clarify.
(10)In the first sentence of this subsection, you indicate that you used a network of process variable sensors and after that, you referred to portable instruments – Are you talking about the same thing here? Please write this subsection entirely to clarify the message you want to convey.
2.2. Selecting measurement points in the cardboard production line
(11)Which process variables are you measuring here? You should list all your process variables and how each of them affects the performance of the cardboard production system to convince your readers why these process variables are important.
(12)You rightfully indicate that these temperatures are measured at these points you indicated. What happens if you don’t measure these temperatures at these points? Please explain that in the text with the supporting references if available as well. I understand you only indicate that the temperature measurement is done directly before the glue units called single facer and double facer or double backer (second paragraph of the introduction – Line 46, page 2), but the places you selected are not consistent with these measurement points or at least is not properly explained what these temperatures should be measured at these points.
3. Results
3.1. Presentation and discussion of results of continuous measurement of process data
(13)Please provide the full information about the software you used – Information about the developer, version, etc. should be indicated. If you developed the software by yourself, then you should also indicate that as well as the programming environment and the features of the computer used to do that.
4. Discussion - Design of the algorithm for the selection of the desired values of process quantities taking into account the individual preferences of the operator
(14)The discussion of your results is descriptive instead of quantitative. That is adequate calculations should be made to indicate the performance of your design to justify why your proposed method is important.
(15)I suggest the authors provide an independent section discussing the results of the experiments separate from the results section. In doing so, please provide a quantitative analysis of your results and outline the comparison performance of the different methods in the paper. Also, mention any difficulties encountered when conducting the research (if any) and what you did to solve them.
A moderate editing of the English and style is required. There is a lot of sentence twisting that can and should improved to ensure the paper is easy to read and understand.
Reviewer 3 Report
In this experimental research it is studied the optimization of the corrugated cardboard production process.
I suggest a more structured abstract includinhg background of the problem.
At the end of the introductory paragraph, I suggest to present the objective of the research.
At the beginning of the Materials and methods paragraph, a brief presentation of the methodological stages would be desirable, after which to move on to the actual detailed presentation.
In paragraph 3.1, the discussion of the results should be moved to the next paragraph.
The discussion section should be improved by reporting on other similar results from the literature. Finally, the limitations of the study should be indicated.
The conclusions section is concise.
Bibliographic references require uniform writing, e.g. 9, 10, etc., uniform indication of doi, authors' names, and many other deviations.
Round 2
Reviewer 2 Report
The authors have responded to all my comments and I believe the content of this paper will be of interest to researchers working on the thematic topic. I have no other questions/concerns and I recommend it for publication in "Sensors".